# Immune Reconstitution Inflammatory Syndrome Associated Kaposi Sarcoma

**DOI:** 10.3390/cancers14040986

**Published:** 2022-02-16

**Authors:** Isabelle Poizot-Martin, Sylvie Brégigeon, Romain Palich, Anne-Geneviève Marcelin, Marc-Antoine Valantin, Caroline Solas, Marianne Veyri, Jean-Philippe Spano, Alain Makinson

**Affiliations:** 1Assistance Publique-Hôpitaux de Marseille (APHM), Inserm, Institut de Recherche pour le Développement (IRD), SESSTIM, Sciences Economiques & Sociales de la Santé & Traitement de l’Information Médicale, ISSPAM, APHM Sainte-Marguerite, Service D’immuno-Hématologie Clinique, Aix-Marseille Université, 13009 Marseille, France; 2Assistance Publique-Hôpitaux de Marseille (APHM) Sainte-Marguerite, Service D’immuno-Hématologie Clinique, Aix-Marseille Université, 13009 Marseille, France; sylvie.ronot@ap-hm.fr; 3Department of Infectious Diseases, Pitié-Salpêtrière Hospital, Assistance Publique-Hôpitaux de Paris (APHP), Pierre Louis Epidemiology and Public Health Institute (iPLESP), INSERM U1136, Sorbonne University, 75013 Paris, France; romain.palich@aphp.fr (R.P.); marc-antoine.valantin@aphp.fr (M.-A.V.); 4INSERM, Institut Pierre Louis d’Epidémiologie et de Santé Publique (iPLESP), AP-HP, Hôpital Pitié Salpêtrière, Service de Virologie, Sorbonne Université, 75013 Paris, France; anne-genevieve.marcelin@aphp.fr; 5Assistance Publique-Hôpitaux de Marseille (APHM), Hôpital La Timone, Laboratoire de Pharmacocinétique et Toxicologie, INSERM 1207, IRD 190, Unité des Virus Emergents, Aix-Marseille Université, 13005 Marseille, France; caroline.solas@ap-hm.fr; 6Department of Medical Oncology, Pitié Salpêtrière Hospital, Assistance Publique-Hôpitaux de Paris (APHP), Institut Universitaire de Cancérologie (IUC), CLIP2 Galilée, Pierre Louis Epidemiology and Public Health Institute (iPLESP), INSERM U1136, Sorbonne Université, 75013 Paris, France; marianne.veyri@aphp.fr (M.V.); jean-philippe.spano@aphp.fr (J.-P.S.); 7Centre Hospitalier Universitaire de Montpellier, Département des Maladies Infectieuses et Tropicales, INSERM U1175/IRD UMI 233, 34000 Montpellier, France; a-makinson@chu-montpellier.fr

**Keywords:** Kaposi sarcoma, immune reconstitution inflammatory syndrome, IRIS, HIV, AIDS, target therapies

## Abstract

**Simple Summary:**

Kaposi sarcoma (KS) incidence has declined substantially since the advent of effective ART but it remains a frequent cancer among people living with HIV, including those on ART with a sustained undetectable HIV viral load and in late presenters. ART is responsible for KS improvement and resolution, but new onset (unmasking KS-IRIS) or sudden progression of preexisting KS (paradoxical KS-IRIS) can occur, even in patients with a low degree of immunocompromise. Both paradoxical and unmasking KS-IRIS have been associated with significant morbidity and mortality. We carried out a literature review regarding the incidence, pathogenic mechanisms, risk factors, clinical presentation, and management strategies of KS-IRIS in the ART era.

**Abstract:**

People living with HIV (PLWH) with advanced immunosuppression who initiate antiretroviral therapy (ART) are susceptible to the occurrence of an immune reconstitution inflammatory syndrome (IRIS). Although ART is responsible for AIDS- associated Kaposi sarcoma (KS) improvement and resolution, new onset (unmasking KS-IRIS) or sudden progression of preexisting KS (paradoxical KS-IRIS) can occur after a time delay of between a few days and 6 months after the initiation or resumption of ART, even in patients with a low degree of immunocompromise. KS-IRIS incidence varies from 2.4% to 39%, depending on study design, populations, and geographic regions. Risk factors for developing KS-IRIS include advanced KS tumor stage (T1), pre-treatment HIV viral load >5 log_10_ copies/mL, detectable pre-treatment plasma-KSHV, and initiation of ART alone without concurrent chemotherapy. Both paradoxical and unmasking KS-IRIS have been associated with significant morbidity and mortality, and thrombocytopenia (<100,000 platelets/mm^3^ at 12 weeks) has been associated with death. KS-IRIS is not to be considered as ART failure, and an ART regimen must be pursued. Systemic chemotherapy for KS in conjunction with ART is recommended and, in contrast with management of IRIS for other opportunistic infections, glucocorticoids are contra-indicated. Despite our preliminary results, the place of targeted therapies in the prevention or treatment of KS-IRIS needs further assessment.

## 1. Introduction

People living with HIV (PLWH) and with advanced immunosuppression who initiate antiretroviral therapy (ART) are susceptible to the occurrence of an immune reconstitution inflammatory syndrome (IRIS) [1]. This syndrome results from the restoration of a dysregulated immune response after ART initiation, optimization or resumption [2], causing an exaggerated inflammatory response against persistent infectious or non-infectious pathogen-specific antigens [1,3,4]. Due to the persistence of a high proportion of late presenters at the time of HIV diagnosis, including in countries with universal access to ART [5], IRIS risk still remains, increasing the burden of morbidity and mortality [1,2].

A wide spectrum of clinical manifestations can be observed during IRIS with two distinct temporal patterns recognized as (1) paradoxical IRIS, when there is a worsening or recurrence of previously treated opportunistic infections (OI) symptoms or other inflammatory auto-immune or neoplastic processes that occurs despite an earlier favorable response to therapy prior to ART, and (2) unmasking IRIS, when one or more of these events occurred following ART initiation [6,7,8,9].

KS has long remained the most common malignant condition in people living with HIV (PLWH) and is one of the most severe complications of full-blown AIDS.

KS is now recognized as a cytokine-mediated angioproliferative disease caused by a human herpesvirus [8] (HHV8), also referred as the Kaposi sarcoma-associated herpesvirus (KSHV) [10,11,12]. KSHV is an oncogenic virus belonging to the *Herpesviridae* family found in the lesions of all epidemiological forms of Kaposi’s sarcoma [13]. KSHV encodes oncogenic proteins that can modulate cellular pathways, leading to inhibition of apoptosis, cell proliferation stimulation, angiogenesis, inflammation, and immune escape, and therefore is involved in KS development [14,15].The widespread use of ART has modified the natural history and incidence of KS as well as its clinical features and prognosis in resource-rich countries more than in low-income countries, particularly in sub-Saharan Africa (SSA) where KS is still considered as a public health challenge [16,17].

In resource-rich countries, KS incidence has declined substantially since the advent of effective ART [18,19], but it remains a frequent cancer among PLWH [20], including those on ART with a sustained undetectable HIV viral load [21,22,23] and in late presenters, as observed between 2010 and 2015 in the French Dat’AIDS cohort [24]. Furthermore, KS is still an important cause of death [25]. Although ART is responsible for KS improvement and resolution, the new onset or sudden progression of preexisting Kaposi sarcoma (KS) within 3 months of the initiation of ART is a well described event among PLWH [26,27,28,29].

We carried out a literature review regarding the incidence, pathogenic mechanisms, risk factors, clinical presentation, and management strategies of KS-IRIS in the ART era.

## 2. Definition of KS-IRIS

General IRIS case definitions, according to French et al. and Shelburne et al., are reported in Table 1 [30,31]. The exaggerated and dysregulated cellular immune responses observed during IRIS could be due to an expansion of regulatory T cells that would not expand at the same rate as the antigen-specific effector cells [32]. Although this pathogen-specific immune response usually results in inflammation of tissues infected with the pathogen, cellular proliferative disease may also occur, explaining Hodgkin-lymphoma, non-Hodgkin lymphoma and KS-IRIS cases [8].

International consensus case definitions have been developed for tuberculosis-IRIS (TB-IRIS) [33] and cryptococcosis-IRIS [34], but not for KS-IRIS; however, there is a need for this [35]. This lack of international consensus case definitions might be explained, at least in part, by the lower incidence rate of KS-IRIS (see below in the incidence paragraph).

Therefore, definitions of KS-IRIS have varied according to different studies. French et al. defines KS-IRIS as either an abrupt clinical worsening of a previously existing KS or a new presentation of a previously unknown KS in temporal association with initiation or reinitiation of ART or change to a more active regimen. For each of these clinical situations, KS-IRIS occurrence is associated with a concomitant reduction of at least 1 log_10_ in the HIV-1 RNA level, or with two of the following three minor criteria: (a) a 2-fold increase in the CD4+ T-cell count after ART, (b) an increase in the immune response (KSHV-antibodies), and (c) a spontaneous resolution of disease without specific chemotherapy with continuation of ART [30].

In the study conducted by Volkow et al., KS-IRIS diagnosis was considered if a clinical abrupt worsening of previously existing KS (“paradoxical”) or the development of KS (“unmasked”) occurred within the first 6 months after the initiation of ART, associated with a reduction of at least 1 log10 of HIV-1 RNA and/or an increase of ≥50 cells/mm^3^ or ≥two fold rise in baseline CD4+ cell count. Furthermore, KS exacerbation was defined as having at least two of the following criteria: abrupt increase in the size or number of KS lesions, the appearance or exacerbation of lymphedema, and the appearance or increase of otherwise unexplained, gallium-negative, lung opacities on chest X-rays after starting ART [36].

Furthermore, in the absence of sufficient viral load reduction, Novak et al. proposed as an adequate response to ART a 50% or greater increase in CD4 cell count within 6 months after initiating a new or different ART regimen [2].

According to Bower et al., the major factors used to diagnose IRIS-associated KS are the temporal relationship between rapid clinical progression of KS and the initiation of ART ([28]). Thus, the lack of a standardized definition of KS-IRIS makes comparisons of incidence rates, risk factors, and the evaluation of therapeutic strategies of KS-IRIS difficult. In our opinion, the definition of KS-IRIS should combine the criteria proposed by Volkow et al. [36] and Novak et al. [2].

There is also a need to distinguish between KS-IRIS and a recurrence/occurrence of KS despite virological control under ART. We and others [16,17,32] have increasingly described cases of KS in virological controlled patients [37]. Most cases occur in well-controlled PLWH, under long-term antiretroviral therapy (and not at the start of ART), and do not usually correspond to the exacerbation of pre-existing Kaposi lesions. Several immunopathological hypotheses could explain such cases, including KSHV chronic antigen exposure, immune modulation by viral proteins, and local immune exhaustion, leading to the consideration of such KS presentation as a specific pattern of KS [22]. However, separating these two entities is essential, as their prognoses and therapeutic strategies differ.

## 3. Pathogenic Mechanisms of KS-IRIS

The pathogenesis of KS involves KSHV, cytokines, and the host-immune response.

In HIV-infected individuals, infection by KSHV is a necessary but not sufficient condition for KS development. Furthermore, not all HIV-KSHV coinfected individuals will develop the disease. Immunosuppression and HIV viral load have been found to be correlated with KS occurrence in PLWH [18,38]. Additionally, CD4:CD8 ratio <1 and CD8 hyperlymphocytosis greater than 1000/mm^3^ regardless of CD4 T-cell count have been found to be associated with an increased risk of KS [18,39].

HIV-1 infection might promote KSHV pathogenesis through the effects of HIV-1 proteins (tat protein) on KSHV replication and indirectly through CD4 lymphocyte depletion and the production of inflammatory cytokines (N-Kappa B activation, INF gamma, VEGF) [11,12,40].

KSHV induces viral and host cytokine production that promotes KS cell proliferation, differentiation and angiogenesis (IL-6, oncostatin, alfa-TNF, PDGF, VEGF) [10,41,42]. Previous studies have confirmed the major role of KSHV-specific T cellular immunity in the control of KS among PLWH [43,44,45,46,47].

Hence, successful control of KS obtained after ART initiation may be attributable to immune restoration and inhibition of HIV replication, thus decreasing HIV-1 angiogenic tat protein concentrations that promote KSHV survival and tumorigenesis within infected cells [48,49], and inhibiting intracellular cytokine production. The reduction of IL-6 limits the interaction between HIV-1 and KSHV and thus the progression of KS [50].

IRIS results from the rapid expansion of antigen-specific CD4+ and CD8+ lymphocytes following initiation of ART, with an imbalance between pro- and anti-inflammatory immune responses [51]. KS-IRIS may be partly explained by the restoration of a KSHV-specific immune T cell response that reduces KSHV replication while promoting paradoxical cellular proliferation and inflammation, leading to KS progression [51,52]. According to Letang et al., in patients with high KSHV dissemination and high HIV viral load, KSHV replication in tissue cells could not be controlled by the restored KSHV-specific immune responses, that could promotes a cytokine-induced reactive angioproliferation and tumorigenesis, resulting in the development of paradoxical KS-IRIS [35].

## 4. Incidence of KS-IRIS and Outcomes

A meta-analysis [53] of cohort studies published between 1998 and 2009 including 13,103 patients living with HIV starting ART reported 1699 (13%) cases of IRIS, of whom 6.4% were KS-IRIS. This rate appears to be much lower than the incidence rate of IRIS of 37.7% observed in patients with CMV retinitis [1].

The incidence of KS-IRIS varies widely, from 2.4% to 39%, depending on study design, populations, geographic regions, and the type of IRIS investigated (paradoxical vs. unmasking *IRIS*) [2,28,35,36,52,53,54,55,56,57]. The paradoxical KS-IRIS incidence in newly diagnosed HIV-positive patients initiating ART was reported to be 6% to11% in three studies from the USA, UK and Mozambique [28,53,54], and that of unmasking KS-IRIS in ART naïve patients, 5% and 4% in Mozambique [54] and a cohort in Mexico, respectively [36]. In two retrospective studies of patients living in the United States, the overall IRIS incidence rate in the first year of ART initiation or change was 11% [2,53]. However, in Achenbach et al.’s study investigating only paradoxical IRIS cases, the KS-IRIS incidence was 29% [53], while in a study performed by Novak et al., in which only unmasking IRIS cases were assessed, the KS-IRIS incidence was 2.4% [2]. Although there are few relevant studies, unmasking KS-IRIS seems to be less frequent compared to paradoxical KS-IRIS. However, some KS disease progression despite ART may have been misclassified as KS-IRIS [53].

KS-IRIS incidence and associated mortality were reported to be higher in low-resource settings compared to resource-rich countries [35]. In a pooled analysis of three prospective cohorts of ART-naïve HIV-infected patients with KS from sub-Saharan Africa and one from the UK, 13.9% of patients experienced paradoxical IRIS-KS, 20% among SSA cohorts and 8.5% in UK, with mortality 3.3-fold higher in the SSA cohorts. Factors predictive of mortality were KS-IRIS, lack of chemotherapy, pre-ART CD4 cell count of less than 200/mm^3^, and detectable baseline KSHV DNA in the plasma [35]. The prevalence of these factors was high in the African cohorts. Furthermore, the rate of KS-IRIS mortality observed in this study was higher than for any other paradoxical IRIS event associated with major opportunistic infections [35].

However, in two urban ART clinics in South Africa [55] with IRIS incidence 22.9%, of which 64% was unmasking-IRIS and 36% paradoxical IRIS, IRIS accounted for 24% of deaths, only one of which was due to paradoxical KS-IRIS [55].

The contribution to mortality of IRIS was also investigated among 2610 PLWH from the HOPS cohort in Washington. In this study, an increased risk of death was found in patients having experienced an IRIS event, with 23.2% of deaths versus 8.5% among those who had not, with a median delay of 40.6 months [2]. Unfortunately, risk of death and time delay to death was not assessed for each type of IRIS but visceral KS-IRIS was found to lead to considerable morbidity and mortality compared with IRIS related to other OIs [53].

## 5. KS-IRIS Associated Risk Factors

Most studies have examined risk factors for paradoxical and unmasking IRIS. and few for KS-IRIS [58,59].

In the study by Achenbach et al., a greater median increase in CD4 T cell count was identified in patients with mucocutaneous KS-IRIS [53]. A higher median increase in CD8 T cell count was also observed in patients with mucocutaneous KS who developed KS-IRIS. Such variations in CD4 and CD8 T cell count were not observed in patients with visceral KS who developed KS-IRIS [53].

Although low CD4 counts at the start of ART drive the incidence of IRIS independently of the pathology involved [1,36], KS-IRIS can occur at higher CD4 cell counts [28,60].

In a prospective cohort of 69 HIV infected adults conducted in Mozambique, Letang et al. identified four independent predictors of KS-IRIS including clinical pretreatment KS, detectable plasma KSHV DNA, hematocrit <30%, and high plasma HIV viral load (≥10 ^5^ Log/mL) [52]. Additionally, in another study, T1 KS stage disease and high pre-ART plasma HIV VL were both reported to be associated with more than twice the risk of developing KS-IRIS, independently of the baseline CD4 cell counts, and KSHV DNA detectability in plasma was associated with a three-fold increase [35].

The association between T1 KS stage and KS-IRIS risk found in this study confirms the hypothesis that higher KS disease burden increases the risk of KS-IRIS, as suggested for IRIS associated with other OI [26,61,62,63]. The higher incidence of KS-IRIS observed in patients with visceral KS compared with patients with mucocutaneous KS reported in a study by Achenbach et al. confirmed these results [53]. Likewise, Volkow et al. reported that the extent and number of KS lesions as well as pulmonary involvement were significantly more in KS-IRIS individuals [36].

However, in a study by Bower et al., neither tumor stage nor site of KS were associated with KS-IRIS risk, while KS-associated edema at baseline was more likely to be observed in patients with KS-IRIS [28]. Likewise, edema interfering with physical function and raised cutaneous lesions before ART initiation were reported to be more frequent in patients who developed KS-IRIS [64].

Of note, concurrent or recent use of glucocorticoids may lead to increased risk of KS-IRIS [65].

Regarding KSHV DNA, detectable baseline plasma KSHV DNA was found to predict KS-IRIS. However, plasma KSHV DNA can be undetectable during KS-IRIS, as previously reported [66].

## 6. Clinical Presentations of KS-IRIS

The time of onset of KS-IRIS is variable, from a few days to 6 months after ART initiation, with the greatest risk within the first 2 months [4,27,28,35]. Haddow et al. have found that unmasking KS-IRIS occurred slightly but significantly later than paradoxical IRIS, with a median 21 days (IQR 7–52 days) and 11 days (IQR 7–24), respectively [55]. A longer median time of onset for mucocutaneous KS-IRIS cases than for other types of IRIS 144 vs. 36 days) has been reported by Achenbach et al., but the difference was not significant [53].

A short onset delay after ART initiation does not seem to have any predictive value, and KS-IRIS can remain a time-limited phenomenon, controlled with a limited course of early systemic chemotherapy [27] or continue to develop, despite several cycles of chemotherapy over a number of months [67].

Depending on the clinical presentation and the location of the lesions, KS-IRIS can be life-threatening but also lead to functional or aesthetic consequences.

Thus, paradoxical KS IRIS frequently presents as inflammation and enlargement of existing KS lesion with marked lesional swelling, increased tenderness, and peripheral edema or worsening edema [68]. It is of note that edema can persist after resolution of KS skin lesions [53].

KS may also extend or appear rapidly at new anatomical sites with variable symptomatology [58]. In its disseminated form, lymph node and visceral involvement can be observed, in particular in the respiratory and gastrointestinal tract, but every organ can potentially be affected, such as bone and cartilage [69] or kidney [70]. In case of gastrointestinal tract involvement, although most often asymptomatic, significant complications may occur, including perforation, bleeding and obstruction [71,72].

Pulmonary manifestations include cough, dyspnea, hemoptysis, parenchymal nodular lesions, adenopathy, and pleural effusions that can necessitate thoracentesis and talc pleurodesis [73]. The presence of endobronchial lesions can lead to acute airway obstruction, which is potentially life-threatening [51,74].

More atypical presentations of KS-IRIS have been reported, such as oral localization [75], chylous ascites and chylothorax [67]. Although rare, KS-IRIS can also be observed in children, as recently reported in a 9-year-old Guinean girl [76].

Diagnosis of KS can be confirmed with immunohistochemical staining using endothelial markers such as CD31, CD34, D2-40, and ERG, through skin biopsy or alternatively endoscopic, pleural ortransbronchial biopsy. No current guidelines recommend the use of virological diagnosis tools to detect the HHV-8 genome or antibodies for KS diagnosis, but HHV-8 can be identified by using LNA-1 antibody or by PCR. Imaging using modalities such as CT, MRI, and FDG-PET/CT scans can aid in diagnosing and staging cutaneous KS and lymph node involvement, as well as responses to treatment of KS lesions [77,78].

Both paradoxical and unmasking KS-IRIS have been associated with significant morbidity and mortality in patients with visceral disease, including persistence of mucocutaneous lesions, lymphedema and death [26,27,79,80,81], sometimes occurring shortly after IRIS diagnosis [53]. Pulmonary involvement is one of the major risks for mortality, with a prevalence reported to be 19 to 35%. [27,36,79,82]. Thrombocytopenia (<100,000 platelets/mm^3^ at 12 weeks) was found to be significantly associated with death [36].

More severe clinical presentations have been reported, with abrupt clinical exacerbation, fever, thrombocytopenia, anemia, hyponatremia, and hypoalbuminemia [83]. However, these symptoms combined with signs of systemic inflammatory syndrome (SIRS) whould suggest the diagnosis of Kaposi Sarcoma Inflammatory Cytokine Syndrome (KICS).This syndrome is associated with a high rate of morbidity and mortality, and its diagnosis requires excisional biopsy of lymphadenopathy to exclude multicentric Castleman disease [84] and facilitate the initiation of prompt treatment [85,86].

## 7. Prevention of KS-IRIS

In contrast with the prevention of TB-IRIS which relies on a delay of ART initiation of at least four weeks after TB treatment initiation [87], ART should not to be delayed in the case of KS. The controversy regarding IRIS risk after ART initiation including integrase inhibitors [88,89], appears to be unfounded, following the results of a recent meta-analysis [90].

The best therapeutic approach for the prevention of KS-IRIS is still not currently defined. Management of AIDS-related Kaposi sarcoma depends on the location and extent of the disease, according to most guidelines [68,91]. Indications for systemic therapy include visceral disease, painful or ulcerated lesions, edema, extensive cutaneous disease, rapidly progressive disease, psychological withdrawal because of the stigma of visible lesions, and KS-IRIS [92]. In other cases, maintaining ART without additional systemic chemotherapy may be sufficient.

Thus, decisions about whether to initiate chemotherapy together with ART to prevent KS-IRIS must consider the extent of the diseaseand the potential benefits and added toxicities of chemotherapy plus ART over ART alone, in particular for individuals with severe immunosuppression. Furthermore, potential drug–drug interactions between ART and chemotherapy need to be assessed [93]. Whatever the decision, close monitoring of patients should be implemented in order to identify the first manifestations of KS-IRIS as quickly as possible.

### 7.1. Chemotherapy for Reduction of KS-IRIS Risk

A meta-analysis of six randomized trials and three observational studies involving 792 HIV-infected patients with severe KS has shown that ART plus chemotherapy with paclitaxel, liposomal doxorubicin, or daunorubicin facilitated a significant reduction of disease progression, with no differences between the three drugs [94], but no statistically significant reduction in KS-IRIS, as had been observed in one UK cohort study [95]. In this cohort, liposomal anthracyclines plus ART were compared to ART alone in 129 patients with T1KS, with the following results: four out of 65 T1 participants in the ART plus liposomal anthracycline group developed KS- IRIS, compared to eight out of 64 participants in the ART only group (RR 0.49; 95% CI 0.16 to 1.55) [95].

However, Letang et al. reported, in a pooled analysis of three prospective cohorts of ART-naïve PLWH with KS from SAA and one from the UK, a positive impact of adjunctive chemotherapy on KS-IRIS risk [35]. This result was consistent with previous studies conducted in South Africa and Zimbabwe, showing better control of KSHV replication and clinical outcomes among KS patients receiving ART and chemotherapy, vs. ART only [43,96,97].

Recently, three randomized clinical trials conducted in SSA and Brazil evaluated different chemotherapy drug regimens for the prevention of KS-IRIS. However, liposomal anthracyclines, the gold standard of care, are not available in these countries, and only non-liposomal doxorubicin, bleomycin, vincristine, and etoposide may be currently administered, even though they are less effective and have worse side-effect profiles in cases of KS [92].

A randomized trial conducted by Mosam et al. in South Africa found that the addition of chemotherapy with intravenous bleomycin/vincristine to ART did not affect rates of KS-IRIS [96]. In a three-arm randomized clinical trial conducted by Krown et al. in SSA countries and Brazil, suspected KS-IRIS occurred infrequently during treatment of advanced and symptomatic KS with ART in combination with each of three chemotherapy regimens tested, as follows: 10% in etoposide, 2% in bleomycin/vincristine, and none in the paclitaxel arm [98].

In a randomized trial evaluating the impact of initiating ART with either immediate or as-needed oral etoposide in PLWH with mild to moderate KS, immediate etoposide with ART provided early but non-durable clinical benefits [99], and reduced the incidence of KS-IRIS, as follows: 7% in the immediate etoposide with ART arm, versus 21% in the ART only arm [64]. In this trial, etoposide was administered as a single 50mg capsule on days 1–7 of each 14 day cycle.

### 7.2. Antiviral Agent in Reduction of KS-IRIS Risk

Valganciclovir has been evaluated in the prevention of KS-IRIS due to the in vitro activity of this antiviral agent against KSHV and epidemiological data showing a reduction of KS incidence when given prophylactically for CMV infection [100,101,102,103,104,105]. The randomized clinical trial conducted by Volkow et al. on 38 ART naïve patients with disseminated KS found a reduction of severe KS-IRIS events and in the number of patients with at least one severe KS-IRIS episode. Valganciclovir treatment was administered at 900 mg BID, initiated four weeks before ART and maintained for 48 weeks [106]. This result remains to be confirmed in a larger number of patients, and the benefits and risks of such a prophylaxis weighed, particularly due to valganciclovir’s haematological toxicities.

## 8. Management of KS-IRIS

KS-IRIS does not indicate the failure of ART or a need for changes in ART regimen [43,58], and systemic chemotherapy for KS in conjunction with ART is recommended [92].

The staging for AIDS-related KS was developed in adults by the AIDS Clinical Trial Group (ACTG) of the National Institutes of Health in the pre-ART era [107] and was revised in 1997 [108]. This staging is based on the extent of the tumor, immune status, and severity of systemic illness (Table 2) (TIS classification). This classification is more often used in research than in clinical practice, where patients are grouped into (1) clinically severe KS needing an immediate fast-acting intervention, or (2) clinically mild to moderate KS, for which patients can be treated with ART alone. However, the definition of severe KS remains debatable, but disease progression can be evaluated according to the ACTG response criteria (Table 3) [107].

Currently, liposomal anthracyclines and paclitaxel are the two reference treatment for patients suffering from extensive and/or rapidly progressive KS, and liposomal anthracyclines are the gold standard of care. However, to our knowledge, the best therapeutic approach for KS-IRIS has not yet yet evaluated in controlled trials.

In contrast with management of IRIS for some opportunistic infections, glucocorticoids may be harmful for KS-IRIS and are contra-indicated, because of the potential for life-threatening KS exacerbation [65,74,109,110]. Therefore, administration of glucocorticoids should be limited to life-threatening conditions for which glucocorticoids are otherwise indicated (i.e., anaphylaxis) [68], on a case-by-case basis after an assessment of risks and benefits, as in a case report [66], and under close monitoring for the early detection of KS lesions.

In recent years, multiple targeted therapies for KS have been evaluated in clinical trials but, to our knowledge, none in patients with KS-IRIS [111,112,113,114,115,116,117,118,119], which means we are unable to define their place in the prevention or treatment of KS-IRIS. Among these targeted therapies, the demonstration of PD-1 expression in KS has led to the evaluation of immune-checkpoint-inhibitors (ICI) as a possible treatment option for KS [120,121]. A systematic review of the safety and efficacy of ICI therapy in HIV-infected patients with advanced-stage cancer reported an incidence of 8.6% of grade 3 or higher immune-related adverse events (irAEs) [122] but, to our knowledge, no cases of IRIS.

Although no cases of irAEs related to anti-PD-1 therapy have been observed in a group of [9] HIV positive patients with KS 119, clinicians should bear in mind that steroids, which are generally used for irAEs [123], should be avoided in patients with active or prior KS, given the potential to cause significant flares and relapses [68]. Likewise, other therapies that suppress B-and T-cell numbers and /or function, such as rituximab are associated with flares of KS and should be used with caution [124].

Moreover, a case of KSHV-associated B-cell lymphoproliferation, possibly attributable to pembrolizumab, has been observed in a heavily pretreated patient with KS and a history of elevated peripheral circulating cell-associated KSHV and KICS; this emphasizes the need for careful monitoring of these patients during treatment with targeted therapies [125].

## 9. Conclusions

Although KS-IRIS is less frequent than other IRIS-related opportunistic infections, close clinical supervision is warranted for patients with KS who are initiating, changing, or resuming ART, as the onset of IRIS-associated KS flare can be observed in as little as three weeks. Particular vigilance is recommended in the case of KS with visceral involvement and peripheral edema, including in those with minimally altered immune status. Clinicians should remain cautious as IRIS-KS is often not readily recognized and may occur in patients resuming ART after a long interruption period. No preventive treatment for KS-IRIS has yet been validated, and glucocorticoids may be harmful for KS-IRIS treatment, therefore they should be limited to life-threatening cases only.

## Figures and Tables

**Table 1 cancers-14-00986-t001:** General IRIS case definitions.

According to French et al., 2004 [24]	
Diagnosis Requires Two Major Criteria (A+B) or Major Criterion (A) Plus Two Minor Criteria to Be Fulfilled	
Major criteria	Minor criteria
(A) Atypical presentation of opportunistic infections or tumors in patients responding to ART	Increase in blood CD4 T-cell count after starting ART
- Localized disease - Exaggerated inflammatory reaction - Atypical inflammatory response in affected tissues - Progressive organ dysfunction or enlargement of pre-existing lesions after definite clinical improvement with pathogen-specific therapy before the initiation of ART and exclusion of treatment toxicity and new alternative diagnoses	Increase in an immune response specific to the relevant pathogen—e.g., delayed type hypersensitivity skin test response to mycobacterial antigens
(B) Decrease in plasma HIV RNA concentration by more than 1 log10 copies per mL	Spontaneous resolution of disease without specific antimicrobial therapy or tumor chemotherapy with continuation of ART
**According to Shelburne et al., 2006** [31]	
**Criteria for IRIS Diagnosis Include:**	
HIV-infected patient Receiving effective ART as evidenced by a decrease in HIV-[1] RNA concentration from baseline or an increase in CD4+ T cells from baseline (may lag behind HIV-1 RNA decrease) Clinical symptoms consistent with inflammatory process Clinical course not consistent with expected course of previously diagnosed opportunistic infection, expected course of newly diagnosed opportunistic infection, or drug toxicity	

**Table 2 cancers-14-00986-t002:** Staging classification for AIDS-related KS (Krown et al., 1989) [107].

Criteria	Good Risk (All of the Following)	Poor Risk (Any of the Following)
Tumor, T	T0: confined to the skin and/or lymph nodes and/or minimal oral disease (non-nodular KS confined to palate)	T1: Tumor-associated edema or ulceration Extensive oral KS Gastrointestinal KS KS in organs other than lymph nodes
Immune system, I ^1^	I0: CD4 T-cell count ≥150/mm^3^	I1: CD4T cell <150/mm^3^
Systemic disease, S	S0: No history of opportunistic infection or thrush No “B symptoms” ^2^ Karnofsky performance status ≥70	S1: History of opportunistic infection and/or thrush “B” symptoms present Karnofsky performance status <70 Other HIV-related illness (eg, neurologic disease, lymphoma)

^1^ I stage has less prognostic value than T or S stages in the presence of ART therapy ^2^ “B” symptoms are unexplained fever, night sweats, >10 percent involuntary weight loss, or diarrhea persisting more than two weeks.

**Table 3 cancers-14-00986-t003:** Disease progression criteria according to ACTG response criteria (Krown et al., 1989) [107].

- An increase of 25% or more in the size of previously existing lesions - The appearance of new lesions or new sites of disease - A change in the character of 25% or more of the skin or oral lesions from macular to plaque-like or nodular - The development of new or increasing tumor- associated edema or effusions

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
