# Peer review of "Immune Reconstitution Inflammatory Syndrome Associated Kaposi Sarcoma"

_cancers, 2022, doi:10.3390/cancers14040986_

Round 1
Reviewer 1 Report
Even in the HARRT era, Kaposi sarcoma is still the major malignancy and cause of death among HIV-infected individuals. KS-IRIS has become complications of KS and AIDS. The manuscript by Poizot-Martin et al reviewed the studies on incidence, risk factors and potential pathogenesis of KS-IRIS. This review is of clinical and biological significance, timely needed, and appreciated. I have some suggestions aiming to strengthen the article.
(1) This manuscript compiles previously and recently published research on KS-IRIS. The article can be stronger and more compelling if authors provide more critical evaluation and their own thoughts for some studies described in the previous publication, rather than just compiling other’s reports.
(2) The manuscript can be better organized in order to approach the KS-IRIS from clinical manifestations to pathogenesis. For example, section 1 (definition of KS-IRIS), section 4 (incidence of KS-IRIS), and section 6 (Clinical presentations of KS-IRIS) should be together or next to each other. Pathogenic mechanism should be after the after these sections.
(3) The pathogenic mechanisms for IRISs of different pathogens may be quite different. Thus, the manuscript does not need to talk about too much on other IRIS except use them as references for defining KS-IRIS.
(4) The section of pathogenic mechanism is week. That is because that we know very little on the pathogenesis of KS-IRIS. There is just some sketchy idea such as unbalanced reconstitution of effector and regulatory T-cells, and exuberant inflammatory response in patients receiving ART. However, authors may still discuss some hypotheses and offer their own view and thoughts.
(5) It would be appreciated if the manuscript pay attention to KS in different locations among KS-IRIS and the outcomes, such as is oral KS more malignant than skin KS and is gastrointestinal KS with hemorrhage associated with higher mortality?
Author Response
Reviewer 1
Even in the HARRT era, Kaposi sarcoma is still the major malignancy and cause of death among HIV-infected individuals. KS-IRIS has become complications of KS and AIDS. The manuscript by Poizot-Martin et al reviewed the studies on incidence, risk factors and potential pathogenesis of KS-IRIS. This review is of clinical and biological significance, timely needed, and appreciated. I have some suggestions aiming to strengthen the article.
- This manuscript compiles previously and recently published research on KS-IRIS. The article can be stronger and more compelling if authors provide more critical evaluation and their own thoughts for some studies described in the previous publication, rather than just compiling other’s reports.
We thank the reviewer for this suggestion. The manuscript has been changed as follow
Line 139: We have added “In our opinion, KS-IRIS definition should combine the criteria proposed by Volkow et al (ref 36) and Novak et al (ref 2).”
Line 145: Several immunopathological hypothesis could explain such cases including KSHV chronic antigen exposure, immune modulation by viral proteins and local immune exhaustion, leading to consider such KS presentation as a specific pattern of KS.
- The manuscript can be better organized in order to approach the KS-IRIS from clinical manifestations to pathogenesis. For example, section 1 (definition of KS-IRIS), section 4 (incidence of KS-IRIS), and section 6 (Clinical presentations of KS-IRIS) should be together or next to each other. Pathogenic mechanism should be after the after these sections.
Answer: We thank the reviewer for this suggestion but we think it is more appropriate to keep the paragraph on etiopathogenetic mechanisms after having defined KS IRIS and then continue with incidence, clinical manifestations and management. We hope this will suit the reviewer.
3) The pathogenic mechanisms for IRISs of different pathogens may be quite different. Thus, the manuscript does not need to talk about too much on other IRIS except use them as references for defining KS-IRIS.
We agree with the reviewer’s opinion. The manuscript has been modified as follows:
Line 224: Most studies have examined risk factors for paradoxical and unmasking IRIS and few for KS-IRIS (57) (58).
Line 226 to 247
5.1. Risk factors for IRIS
According to Walker et al, risk factors for HIV- associated IRIS in PLWH can be divided in host-related risk factors, pathogen-related risk factors and treatment-related risk factors and are reported in the table 4 (57). Several host-based markers, including genetic markers, immune metabolomic and inflammatory biomarkers have been studied to improve predictive and diagnostic tools for IRIS in PLWH who have a concomitant OI but, at our knowledge, none for KS-IRIS (58). According to these studies, host biomarkers of immune activation emerge as an option to identify patients at increased risk of IRIS development.
Table 4. Risk factors for HIV-associated IRIS.
|
|
Risk factor |
|
Host-related |
Low CD4 cell count at ART initiation Opportunistic infection or Tuberculosis prior to ART initiation Genetic predisposition (eg, HLA-A,-B44,-DR4(associated with herpes virus IRIS; TNFA-308*I,IL6-174*G (associated with mycobacterial IRIS) Paucity of immune response at OI diagnosis (in the case of C-IRIS) |
|
Pathogen-related |
Degree of dissemination of OI/burden of infection (eg TB, KS, cryptococcosis) High pre-ART HIV viral load |
|
Treatment-related |
Shorter duration of OI treatment prior to starting ART (paradoxical IRIS) Rapid suppression of HIV viral load |
|
C-IRIS, cryptococcal-associated IRIS;TB, tuberculosis; KS, Kaposi sarcoma |
|
Risk factor profile has been reported by Haddow et al to differ according to IRIS pattern, the baseline HIV viral load and the duration of OI therapy prior to ART being associated with paradoxical OIs-IRIS while pre-ART lymph nodes, haemoglobin level, C-reactive protein rate and a weight loss prior to ART were associated with unmasking OIs-IRIS(54). In this study, magnitude and rate of CD4 immune reconstitution and VL reduction following ART initiation were not predictive of any type of IRIS while In the study by Achenbach et al, a greater median increase in CD4 T cell count has been identified in patients with mucocutaneous KS-IRIS and tuberculosis-IRIS (52). A higher median increase in CD8 T cell count was also observed in patients with mucocutaneous KS who developed KS-IRIS but not in patients with tuberculosis-IRIS. Such variations in CD4 and CD8 T cell count were not observed in patients with visceral KS who developed KS-IRIS (52).
3) The section of pathogenic mechanism is week. That is because that we know very little on the pathogenesis of KS-IRIS. There is just some sketchy idea such as unbalanced reconstitution of effector and regulatory T-cells, and exuberant inflammatory response in patients receiving ART. However, authors may still discuss some hypotheses and offer their own view and thoughts.
Answer: We thank the reviewer for this remark. Unfortunately, we cannot argue on new etiopathogenetic hypotheses as we have not conducted specific studies on this topic. Furthermore, for this manuscript, our objective was to carry a literature review in order to raise awareness of clinicians on KS-IRIS.
The text has been modified Line 154: The pathogenesis of KS involves KSHV human herpes virus 8 (HHV8) also referred as KS-associated herpes virus (KSHV), cytokines and host-immune response.
(4) It would be appreciated if the manuscript pay attention to KS in different locations among KS-IRIS and the outcomes, such as is oral KS more malignant than skin KS and is gastrointestinal KS with hemorrhage associated with higher mortality?
Answer: we thanks the reviewer for drawing our attention to the fact that these points have not been sufficiently highlighted in our article although addressed overall in paragraphs 4 ‘Incidence and outcomes’ and according to clinical presentation in paragraph 6. ‘Clinical presentation of KS-IRIS’.
Line 298: In case of gastrointestinal tract involvement, significant bleeding may occur. Pulmonary manifestations include cough, dyspnea, hemoptysis, parenchymal nodular lesions, adenopathy and pleural effusions that can necessitate thoracenteses and talc pleurodesis(70). The presence of endobronchial lesions might lead to acute airway obstruction, potentially life-threatening (50,71).
Line 316: Both paradoxical and unmasking KS-IRIS have been associated with significant morbidity and mortality in patients with visceral disease including persistent of mucocutaneous lesions, lymphedema and death (20,21,74–76), sometimes occurring shortly after IRIS diagnosis (52). Pulmonary involvement is one of the major risk for mortality with a prevalence reported to be 19 to 35% (21,31,74,77). Thrombocytopenia less than <100,000 platelets/mm3 at 12 weeks was found to be significantly associated with death (31).
We did not found studies evaluating mortality according to clinical presentation as explained in the manuscript Line 219:
Line 219: Unfortunately, risk of death as time delay to death was not assessed for each type of IRIS but visceral KS-IRIS was found to lead to considerable morbidity and mortality compared with IRIS -related to others OIs (52).
The manuscript has been modified as follows Line 288 and in Line 290:
“Depending on the clinical presentation and the location of the lesions, KS-IRIS can be life-threatening but also leave functional or aesthetic sequelae.
Thus, paradoxical KS IRIS frequently presents with inflammation and enlargement of existing KS lesion with marked lesional swelling, increased tenderness, and peripheral oedema or worsening oedema (69). Of note, oedema can persist after resolution of KS skin lesions (54).”
Line 297 “….such as bone and articulation (70) or kidney (71).”
Line 298: In case of gastrointestinal tract involvement, although most often asymptomatic, significant complications may occur, including perforation, bleeding and obstruction (72, 73).
Line 306: ascites and chylothorax (68).Although rare, KS-IRIS might also be observed in children as recently reported in a 9-year-old Guinean girl (77).
And we added two references
Lee, A.J., et al., Gastrointestinal Kaposi’s sarcoma: Case report and review of the literature. World J Gastrointest Pharmacol Ther,
- 6(3): p. 89-95.
Arora, M. and E.M. Goldberg, Kaposi sarcoma involving the gastrointestinal tract. Gastroenterol Hepatol (N Y), 2010. 6(7): p.
459-62.

Reviewer 2 Report
Since the review about KS-related deregulation of immune responses, which is caused by KSHV infection, I recommend that the authors briefly describe what KSHV and Kaposi’s sarcoma (KS) is before talking about KS-related other diseases.
Improve Table 1, use columns with bullet points to briefly summarizes the criteria
Line 96-103. It is difficult to follow. Break the paragraph up to several sentences.
Table 2. The use of numbers after T, I, and S is confusing. I do not understand what purpose they serve.
Information presented in Table 3 and 5 does not require Table format to my opinion. They are already listed in the main text.
Line 106-112 very long sentence, it needs to be split into shorter sentences. It is difficult to follow what they want to say.
There are a number of very long sentences (over 7-10 lines) throughout the review, which read like lists. The major concern is that the long lists of clinical symptoms along with the listing of clinical/epidemiological studies make very difficult for the readers to focus on what the authors want to say.
Author Response
Reviewer 2
- Since the review about KS-related deregulation of immune responses, which is caused by KSHV infection, I recommend that the authors briefly describe what KSHV and Kaposi’s sarcoma (KS) is before talking about KS-related other diseases.
Answer: We thanks the reviewer for this suggestion. The manuscript has been modified as follows in the paragraph Introduction:
Line 74: KS is now recognized as a cytokine-mediated angioproliferative disease caused by human herpes-virus 8 (HHV8) also referred as Kaposi sarcoma-associated herpesvirus (KSHV)(10-12). KSHV is an oncogenic virus belonging to the Herpesviridae family found in the lesions of all epidemiological forms of Kaposi’s sarcoma (13). KSHV encodes oncogenic proteins which can modulate cellular pathways, leading to inhibition of apoptosis, cells proliferation stimulation, angiogenesis, inflammation and immune escape and therefore is involved in KS development (14,15).
And we have added three references:
Mesri, E.A.; Feitelson, M.A.; Munger, K. Human Viral Oncogenesis: A Cancer Hallmarks Analysis. Cell Host Microbe 2014, 15, 266–282, doi:10.1016/j.chom.2014.02.011.
Boshoff, C.; Chang, Y. Kaposi’s Sarcoma-Associated Herpesvirus: A New DNA Tumor Virus. Annu. Rev. Med. 2001, 52, 453–470, doi:10.1146/annurev.med.52.1.453.
Jha, H.C.; Banerjee, S.; Robertson, E.S. The Role of Gammaherpesviruses in Cancer Pathogenesis. Pathog. Basel Switz. 2016, 5, E18, doi:10.3390/pathogens5010018.
- Improve Table 1, use columns with bullet points to briefly summarizes the criteria
Answer: Table 1 has been changed.
- Line 96-103. It is difficult to follow. Break the paragraph up to several sentences.
Anwer: We apologize. We hope that the change will made the text more explicit/
Likewise, The staging for AIDS-related KS has been developed in adults by the AIDS Clinical Trial Group (ACTG) of the National Institutes of Health dates from the pre-ART era (29) and was revised in 1997 (30). This staging is based on the extent of tumour, immune status and severity of systemic illness (table 2) (TIS classification). This classification is most used in research than in clinical practice where patients are grouped into (i) clinically severe KS needing an immediate fast-acting intervention or (ii) clinically mild to moderate KS, for which patients can be treated with ART alone. However, definition of severe KS remains still a debate but disease progression can be evaluated according to ACTG response criteria (Table 3)(29).
- Table 2. The use of numbers after T, I, and S is confusing. I do not understand what purpose they serve.
Answer: The reviewer is right. Table 2 and table 3 have been moved to the paragraph 8. Management of KS IRIS with the following modification Line 396:
Likewise, the The staging for AIDS-related KS has been developed in adults by the AIDS Clinical Trial Group (ACTG) of the National Institutes of Health dates from in the pre-ART era (29) and was revised in 1997 (30),….
- Information presented in Table 3 and 5 does not require Table format to my opinion. They are already listed in the main text.
Answer: The table 5 has been suppressed. We think that the table 3 that is now in the paragraph 8 could be maintained.
6) Line 106-112 very long sentence, it needs to be split into shorter sentences. It is difficult to follow what they want to say.
Answer: The reviewer is right. We modified the sentence as follow
Line 114: The criteria proposed by French et al. define KS-IRIS as either an abrupt clinical worsening of a previously existing KS or a new presentation of a previously unknown KS in temporal association with initiation or reinitiation of ART or change to a more active regimen. For each of these clinical situations, KS-IRIS occurrence is associated with a concomitant reduction of at least 1 log10 in the HIV-1 RNA levels at the time of the IRIS event or with 2 two of the following 3 three minor criteria: (a) a 2-fold increase in the CD4+ T-cell count after ART, (b) an increase in the immune response (KSHV- antibodies), and (c) a spontaneous resolution of disease without specific chemotherapy with continuation of ART (24).
7) There are a number of very long sentences (over 7-10 lines) throughout the review, which read like lists. The major concern is that the long lists of clinical symptoms along with the listing of clinical/epidemiological studies make very difficult for the readers to focus on what the authors want to say.
As recommended by the reviewer, we have reduced long sentences throughout the manuscript (please see revised manuscript).

Reviewer 3 Report
POIZOT-MARTIN et al. 's manuscript focuses on KS-IRIS in PLWH, and attempts to provide a review on the phenomenon as well as diagnosis and therapy approaches.
The manuscript overall is well organized and well written with a focus on the clinical audience.
However there are certain concepts that have not been well presented and in the opinion of the reviewer would add more accuracy to the current manuscript.
- Although use of tables to classify different definitions of KS-IRIS is a good idea but the body of the paragraph would benefit from providing more background to what is referred to often as exuberant response to residual pathogens.
- There needs to be more clarification / discussion on the challenges faced in proper diagnosis of KS-IRIS from recurring KS which is currently one of the major pitfalls in KS-IRIS.
- Additionally the manuscript would gain major benefit from discussing needs for diagnostic assays and approached taken in the past few years to better address this challenge such as lung involvement in IRIS patients and ...
Author Response
Reviewer 3
POIZOT-MARTIN et al. 's manuscript focuses on KS-IRIS in PLWH, and attempts to provide a review on the phenomenon as well as diagnosis and therapy approaches.
The manuscript overall is well organized and well written with a focus on the clinical audience.
However there are certain concepts that have not been well presented and in the opinion of the reviewer would add more accuracy to the current manuscript.
- Although use of tables to classify different definitions of KS-IRIS is a good idea but the body of the paragraph would benefit from providing more background to what is referred to often as exuberant response to residual pathogens.
Answer: We thanks the reviewer for this suggestion. The manuscript has been changed as follows
Line 96: The exaggerated and dysregulated cellular immune responses observed during IRIS could be due to an expansion of regulatory T cells that would not at the same rate that the antigen-specific effector cells (32) Although this pathogen-specific immune response usually results in inflammation of tissues infected with the pathogen, cellular proliferative disease may also occur explaining Hodgkin-, non Hodgkin lymphoma- and KS-IRIS cases (8).
And we added one reference
Ruhwald M, Ravn P. Immune reconstitution syndrome in tuberculosis and HIV-co-infected patients: Th1 explosion or cytokine storm? AIDS. 23 avr 2007;21(7):882‑4.
- There needs to be more clarification / discussion on the challenges faced in proper diagnosis of KS-IRIS from recurring KS which is currently one of the major pitfalls in KS-IRIS.
Answer: we thanks the reviewer for drawing our attention to the fact that this point has not been sufficiently highlighted in our article although addressed in Line 141.
The manuscript has been modified as follow:
Line 141 to 149: There is also a need to distinguish between KS-IRIS and recurrence/occurrence of KS despite virological control under ART. We and others (16,17,32) have increasingly described cases of KS in virological controlled patients (33). Most cases occur in well controlled PLWH, under long term antiretroviral therapy (and not at the initial of ART), and do not usually correspond to the exacerbation of pre-existing Kaposi lesions. Several immunopathological hypothesis could explain such cases including KSHV chronic antigen exposure, immune modulation by viral proteins and local immune exhaustion, leading to consider such KS presentation as a specific pattern of KS (22).However, dissociating these two entities is essential, as prognosis and therapeutic strategies differ.
- Additionally the manuscript would gain major benefit from discussing needs for diagnostic assays and approached taken in the past few years to better addressthis challenge such as lung involvement in IRIS patients and ...
Answer: We thanks the reviewer for this suggestion. The manuscript has been modified as follows Line 308
Diagnosis of KS can be confirmed with immunohistochemical staining using endothelial markers such as CD31, CD34, D2-40 and ERG through skin biopsy or alternatively endoscopic, pleural, transbronchial biopsy. No current guidelines recommend the use of virological diagnosis tools to detect the HHV-8 genome or antibodies for KS diagnosis but HHV-8 can be identified by using LNA-1 antibody or by PCR. Imaging using modalities such as CT, MRI, and FDG-PET/CT scans can aid in diagnosing, staging cutaneous KS and lymph node involvement, as well as treatment response of KS lesions (78,79).
And we have added two references:
- Addula D, Das CJ, Kundra V. Imaging of Kaposi sarcoma. Abdom Radiol (NY). nov 2021;46(11):5297‑306.
- Dupin N, Jary A, Boussouar S, Syrykh C, Gandjbakhche A, Bergeret S, et al. Current and Future Tools for Diagnosis of Kaposi’s Sarcoma. Cancers (Basel). 25 nov 2021;13(23):5927.

Round 2
Reviewer 1 Report
The manuscript has been improved after author's modification. I think this review article is appropriate to be published in Cancers.
Reviewer 2 Report
I acknowledge the authors' revision.